# COVID-19 vaccination uptake in 441 socially and ethnically diverse pregnant women

Fatima Husain[1,☯,‡], Veronica R. Powys[1,☯,‡], Eleanor White[1], Roxanne Jones[1], Lucy P. Goldsmith[2], Paul T. Heath[3], Pippa Oakeshott[2], Mohammad Sharif Razai [2]*

1 Department of Obstetrics and Gynaecology, Wexham Park Hospital, Frimley Health Foundation Trust, Frimley, United Kingdom, 2 Population Health Research Institute, St George's, University of London, London, United Kingdom, 3 Vaccine Institute, St. George's, University of London, London, United Kingdom

☯ These authors contributed equally to this work.
‡ FH and VRP are share first authorship on this work.
* mrazai@sgul.ac.uk

## Abstract

### Objective

To explore COVID-19 vaccination uptake, facilitators and barriers in ethnically-diverse pregnant women.

### Design and setting

An anonymous quality improvement questionnaire survey exploring COVID-19 vaccination uptake, causes of vaccine hesitancy and trusted sources of information among pregnant women in two acute district general hospitals in England (Berkshire and Surrey) between 1.9.21 and 28.2.22.

### Population

441 pregnant women attending routine antenatal clinic appointments.

### Methods

Consented pregnant women completed the survey either electronically using a QR code or on paper. Descriptive data were summarised and free text responses were thematically analysed.

### Results

441 pregnant women, mean age 32 years (range 17–44), completed the survey. Twenty-six percent were from ethnic minority groups, and 31% had a co-morbid health condition. Most respondents (66.2%) had been vaccinated against COVID-19 with at least one dose (White British 71.9%, Asian 67.9%, White-other 63.6%, Black 33%). The most common reasons for not being vaccinated were concerns about effects on the unborn baby and future pregnancies, anxiety about possible adverse impact on the mother, not enough known about the vaccine, and lack of trust in vaccines. Comments included: "I'd rather not risk injecting the unknown into my body", and "I don't trust it." Although 23% used social media for information

**Data Availability Statement:** St George's University of London's information governance team has advised that in line with PLOS ONE's guidelines for sharing data, all data relevant to the

study are available within the paper and its supporting information. However, raw data cannot be made publicly available due to the data containing information that could compromise participant privacy and consent.

**Funding:** The authors received no specific funding for this work.

**Competing interests:** The authors have declared that no competing interests exist.

on COVID-19 vaccination, the most trusted sources were the patient's GP and midwife (43%) and official health-related websites such as NHS (39%).

## Conclusions

A third of these pregnant women had not been vaccinated against COVID-19. Trusted health professionals like midwives and GPs could have a crucial role in increasing vaccination uptake.

## Introduction

Pregnant women are at greater risk of severe COVID-19 disease compared to their non-pregnant counterparts [1–3]. Rates of admission to intensive care are three times greater in pregnant women compared to non-pregnant women, with a 25% greater likelihood of death [4]. The risk is higher in pregnant women with health conditions and complications during pregnancy [1, 2, 5]. In addition, women who contract COVID-19 during pregnancy have a higher risk of pre-eclampsia, preterm birth, stillbirth and early neonatal death [5]. Whilst initial advice from the UK's Joint Committee on Vaccination and Immunisation (JCVI) stated in December 2020 there was insufficient evidence to vaccinate during pregnancy routinely [6], it is now known that COVID-19 vaccination during pregnancy can reduce the risk of severe infection and resultant complications [7, 8].

In April 2021, the JCVI made a specific recommendation that pregnant women should be offered a COVID-19 vaccine [8]. To date, observational data have not identified any safety issues with COVID-19 vaccination in pregnancy [9], with similar side-effects reported between vaccinated and non-vaccinated pregnant women [7, 9–11]. However, COVID-19 vaccine uptake and intent to vaccinate remain low amongst pregnant women [3, 5, 7].

A recent UK cohort study found that less than one-third of 1328 eligible pregnant women accepted the COVID-19 vaccination [7]. A prospective cohort study of 131,000 pregnant women in Scotland suggested that less than half of those who gave birth in October 2021 had received any COVID-19 vaccination, and less than a third were fully vaccinated [5]. These vaccination rates are significantly lower than that in the general population, where 77% of non-pregnant women of child-bearing age had been vaccinated with two doses over the same period [5]. Additionally, high rates of vaccine hesitancy [12] (a delay or refusal of safe vaccines despite availability of vaccine services) have resulted in adverse consequences in pregnancy [1]. Data from England show that between February and September 2021, over 98% of 742 pregnant women admitted to hospital with symptomatic COVID-19 were unvaccinated [1]. Even more recent data from November 2021 to January 2022 show about 46% (n = 56,461) of pregnant women remained unvaccinated at delivery, with the highest number of unvaccinated in Black (69.5%, n = 4,164) and mixed ethnicities (56%, n = 1694) [13]. A further study found hospitalisation (n = 748/823, 91%) and intensive care admissions (n = 102/104, 98%) due to COVID-19 were disproportionately high among unvaccinated pregnant women [5].

Vaccine hesitancy is higher in women who are younger [7], have higher levels of socio-economic deprivation [3, 7] and are of Afro-Caribbean or Asian ethnicities [3, 7, 14].

These ethnic groups have also been found to be at greater risk of COVID-19 associated morbidity and mortality [2, 15, 16]. Recent studies have shown reasons for vaccine hesitancy include poor communication of vaccine safety and benefits, lack of recommendations from trusted sources, lack of long-term safety data and mistrust of healthcare providers [17–21].

However, local factors that lead to low vaccine uptake amongst ethnically diverse pregnant women are not well known.

This quality improvement survey aimed to explore uptake, facilitators, and barriers to COVID-19 vaccination amongst pregnant women presenting to the antenatal units of two district general hospitals in Berkshire and Surrey and provide recommendations for improving vaccine uptake.

## Methods

### Survey design

Pregnant women booked for antenatal clinics at Wexham Park and Frimley Park Hospitals (District General Hospitals of Frimley Health NHS Foundation Trust) were invited to participate in an anonymous, voluntary quality improvement survey between 1 September 2021 and 28 February 2022. The two suburban hospitals serve a diverse population group in terms of socio-economic status and ethnicity. The survey was conducted in English. The survey instrument was initially developed based on our current understanding of the influencing factors in COVID-19 vaccination [17, 18]. The final multiple-choice survey questionnaire was modified based on a pilot in five pregnant women (**S1 Fig**).

Participants were asked to provide demographic data on age, ethnicity, main spoken language, parity, gestation, and pre-existing health conditions. Participants were also asked about previous COVID-19 infection and their COVID-19 vaccination status. Those who had not received a COVID-19 vaccine were asked the reasons (barriers). We also asked participants to identify where they received information about COVID-19 vaccines and which single source of information they trusted the most (facilitators). Participants were also asked open-ended questions about COVID-19 vaccination.

According to Frimley Health NHS Trust, this work met the criteria for operational improvement activities exempt from ethics review. (We also applied the Medical Research Council (MRC) Criteria.) Accordingly, it was registered as a Quality Improvement Project (QIP FXP-49) with the Frimley Excellence. Participants were given information about the study and advised that they could choose whether or not to take part without affecting their treatment, and that completing the survey denoted consent to participate.

### Recruitment

Pregnant women presenting for routine appointments at antenatal clinics of the two hospitals were invited to participate. Pregnant women were given information about the survey and encouraged to ask questions. They were provided with a paper copy of the survey by the clinic receptionist or clinical staff (consultant, specialty registrar or midwife) or directed to a QR code link where they could complete the survey online using their smartphone. Additionally, posters advertising the survey with the QR code were placed in the antenatal waiting room and toilet facilities. Most women had a relatively high-risk pregnancy, defined as anyone referred to secondary care for specialist management due to underlying health conditions, maternal age, previous pregnancy-related complications, and other risk factors [22].

### Data collection and analysis

Online versions were collected on Microsoft Forms (MS Forms). Paper versions of the survey were manually inputted into MS Forms, and the data was collated on Microsoft Excel Version 16.58. Statistical analysis was performed by (LPG) using Stata 15.1 (Stata Corp (2017) Stata Statistical Software: Release 15. College Station, TX.) Descriptive data were summarised. The

Pearson Chi-squared test excluding groups with expected counts of less than five was used to test for differences in categorical data.

Analysis of patient free text responses was informed by thematic analysis, conducted by MSR and approved by other team members.

## Results

### Characteristics of participants

There were 441 respondents to the survey in Frimley Park and Wexham Park hospitals (212 and 229 participants, respectively). Due to the nature of the survey, an overall response rate could not be calculated. However, the estimated response rate in women who were actively invited by the healthcare staff in Wexham Park Hospital was 63% (n = 190/300), with a further 39 women completing the survey in response to posters about the survey in clinics. A similar estimated response was received in Frimley Park Hospital. **Table 1** presents the baseline characteristics of the participants. Their mean age was 32 years (range 17–44). Most survey respondents were White British 63.6% (n = 271) and 85.4% (n = 364) of respondents reported English as their main language. About 12% of participants (n = 52) reported a different main language, including Urdu, Punjabi, Pushto, Somali, Swahili, Romanian, Arabic, Amharic, Portuguese, Spanish, Italian and French.

Most participants (n = 299, 68.7%) did not report any underlying health condition. Asthma was the most commonly reported condition (8.7%) followed by diabetes (4.1%) and hypertension (2%). Other conditions included arthritis, idiopathic intracranial hypertension, chronic fatigue syndrome, polycystic ovary syndrome, Crohn's disease, cystic fibrosis, eczema, alopecia, endometriosis, epilepsy, migraines, factor V Leiden, fibromyalgia, lupus, hypothyroidism, irritable bowel syndrome, multiple sclerosis, stroke, ulcerative colitis, and joint hypermobility syndrome.

Most women either reported nulliparity or one previous delivery (n = 306, 76.5%) and most survey participants were in their third trimester of pregnancy (n = 222, 66.2%). Fourteen percent of women (n = 60) reported having had COVID-19, predominantly during the second wave of the current pandemic (September 2020 –April 2021) [23]. A third of these (n = 20) were pregnant at the time of infection.

### COVID-19 vaccination uptake

Most survey respondents (n = 292, 66.2%) chose to have a COVID-19 vaccine. Of those vaccinated, 87.6% received two doses, and most were Pfizer-BioNTech vaccine (75.3%) (**Table 2**). Respondents not receiving the vaccine were most concerned about the effect of the vaccine on

**Table 1. Demographic and clinical characteristics of 441 participants surveyed.**

| Characteristics | Value |
|---|---|
| Age in years (n = 387), mean (range) | 32.0 (17–44) |
| **Ethnicity (n = 426)** | |
| Black African/Caribbean/Black British | 9 (2.1%) |
| White British | 271 (63.6%) |
| White Other | 44 (10.3%) |
| Asian/Asian British | 77 (18.1%) |
| Mixed/multiple ethnic group | 9 (2.1%) |
| Other ethnic group | 4 (0.7%) |

*(Continued)*

**Table 1.** (Continued)

| Characteristics | Value |
|---|---|
| Not disclosed | 12 (2.8%) |
| **English as main language (n = 426)** | |
| Yes | 364 (85.4%) |
| No | 52 (12.2%) |
| Not disclosed | 10 (2.3%) |
| **Comorbidities (n = 435)** | |
| Hypertension | 9 (2.0%) |
| Diabetes | 18 (4.1%) |
| Asthma | 38 (8.7%) |
| Heart condition | 2 (0.5%) |
| Kidney disease | 3 (0.7%) |
| None | 299 (68.7%) |
| Other | 66 (15.2%) |
| **Gestation (n = 375)** | |
| First Trimester | 19 (5.1%) |
| Second Trimester | 108 (28.8%) |
| Third Trimester | 248 (66.1%) |
| **Parity (n = 400)** | |
| 0 | 155 (38.8%) |
| 1 | 151 (37.8%) |
| 2 | 71 (17.8%) |
| 3 | 20 (5.0%) |
| 4 or more | 3 (0.8%) |
| **Previous COVID-19 infection (n = 441)** | |
| Yes | 60 (13.6%) |
| No | 377 (85.5%) |
| Don't know | 4 (0.9%) |
| **Time of positive COVID-19 infection (n = 60)** | |
| First wave dates | 14 (23.3%) |
| Second wave dates | 30 (50%) |
| Third wave dates | 16 (26.7%) |
| **Pregnancy status during previous COVID-19 infection (n = 60)** | |
| Yes | 20 (33.3%) |
| No | 40 (66.7%) |
| **COVID-19 vaccination status (n = 441)** | |
| Vaccinated | 292 (66.2%) |
| Declined | 142 (32.2%) |
| Advised against vaccination | 7 (1.6%) |
| **COVID-19 vaccine received (n = 291)** | |
| AstraZeneca | 55 (19.0%) |
| Pfizer | 219 (75.3%) |
| Moderna | 14 (4.8%) |
| Do not know | 3 (1.0%) |
| **Number of doses of COVID-19 vaccine received (n = 291)** | |
| First dose | 35 (12.0%) |
| Second dose | 255 (87.6%) |
| Do not know | 1 (0.3%) |

**Table 2. COVID-19 vaccination uptake and sources of information in pregnant women.**

| Ethnicity (n = 425) | COVID-19 vaccine uptake n (%) | |
|---|---|---|
| | Accepted | Decline |
| Asian or Asian British | 53 (67.9) | 25 (32.1) |
| Black/African/Caribbean/Black British | 3 (33.33) | 6 (66.67) |
| Mixed/Multiple ethnic groups | 5 (55.6) | 4 (44.4) |
| White British | 194 (71.9) | 76 (28.1) |
| White Other | 28 (63.6) | 16 (36.4) |
| Other ethnic group | 2 (66.67) | 1 (33.33) |
| Not disclosed | 4 (33.3) | 8 (66.7) |
| **Pearson $\chi^2$ test** | $\chi^2$(5 n = 422) = 4.63, p<0.46 | |
| **Reasons for refusing COVID-19 vaccines (n = 142)*** | | |
| Concerns about baby/future pregnancies | 114 (80.3%) | |
| Lack of data on vaccines | 86 (60.6%) | |
| Concerns about vaccine safety for the mother | 63 (44.4%) | |
| Concerns about fertility | 44 (31.0%) | |
| Concerns about breastfeeding | 32 (22.5%) | |
| Previously had COVID-19 infection | 11 (7.7%) | |
| Midwife/doctor did not offer vaccines | 3 (2.1%) | |
| Other | 6 (4.2%) | |
| **Sources of information about COVID-19 vaccines (n = 425)*** | | |
| GP/Midwife | 259(60.9%) | |
| Health-related websites (e,g, NHS) | 248 (58.4%) | |
| Mainstream News organisations | 198 (46.6%) | |
| Friends and family | 153 (36.0%) | |
| Social media (e.g., Facebook, Instagram, YouTube) | 96 (22.6%) | |
| **Single trusted source of information for COVID-19 vaccines (n = 435)** | | |
| GP/Midwife | 186 (42.8%) | |
| Health-related websites | 170 (39.1%) | |
| Other | 45 (10.3%) | |
| Mainstream News Organisations | 15 (3.4%) | |
| Friends and family | 14 (3.2%) | |
| Social Media | 5 (1.1%) | |

*Multiple answers allowed; thus, percentages can sum to greater than 100.

the baby and future pregnancies. In addition, they felt that there was insufficient information known about the vaccine to make a decision (**Table 2** and **Fig 1**).

Vaccination uptake was 72% in White British women, 68% in Asian or Asian British women and 64% in White Other (**Table 2**). Only 33% of Black/African/Caribbean/Black British women were vaccinated; however, numbers were small. Differences were not statistically significant ($\chi^2$(5 n = 422) = 4.63, p<0.46).

## Reasons for COVID-19 vaccine hesitancy from free text responses

Fifty-one participants cited further reasons for refusal to take up the COVID-19 vaccine in the free text box (**S1 Fig**). Thematic analysis revealed the following themes:

**Concerns about the safety and long-term effects of COVID-19 vaccines.** Nearly all who responded to the question mentioned concerns about vaccine safety, side effects, and unknown long-term effects on the health of the unborn baby and/or mother. Most participants

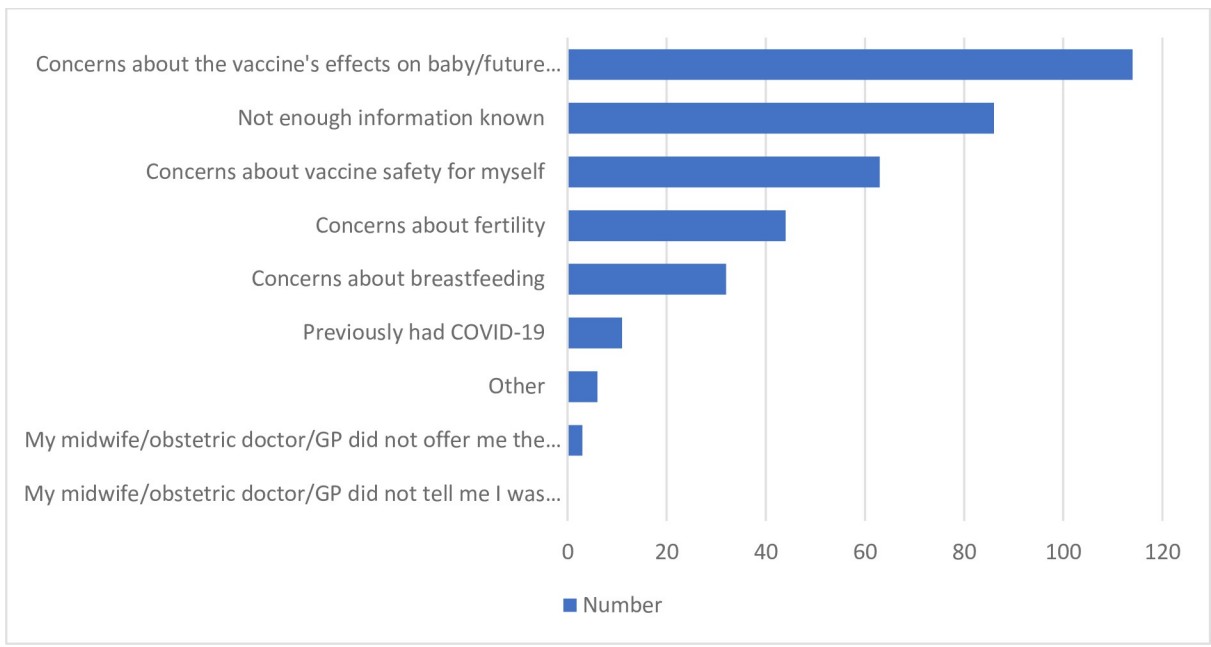

**Fig 1. Pregnant women's reasons for declining COVID-19 vaccination n = 359.**

were worried about the effect of the vaccine on the foetus and its development. Some said they would consider having the vaccine once the baby was born but not whilst they were pregnant.

*"There is no actual facts on how safe this vaccine is during pregnancy. One minute your told pregnant women should not take it then they should. Who knows how safe it really is. Issues will arise in time to come."* 29 years, White Other

*"We have no idea what the long term consequences of this vaccine will be for mother or baby. This is a trial drug."* 39 years, White British

*"I'm concerned mainly about the impact on the baby currently when pregnant. Don't know long term impact."* 33 years, Asian or Asian British

**Lack of information on vaccines' safety in pregnancy.** Most respondents mentioned a lack of data and information on the effect of the vaccine during pregnancy as the reason for their refusal. Further, they believed that the vaccine has not been around for long enough to have an adequate understanding of its safety and long term effects in pregnancy.

*"No proof that the vaccine is safe for the newborn—it hasn't been out long enough. I strongly believe it shouldn't be advertised as safe as they don't know that yet!"* 33 years, White British

*"There is no long term data to show any adverse effects on a foetus or adult. The unvaccinated are being told that they must be vaccinated to be protected but the vaccinated are being told that they are not fully protected and need a "booster". For this reason, at this moment in time I choose not to have the vaccine as I have no faith."* 25 years, White British

*"It is still a new vaccine with limited data on side effects and if it works."* 34 years, Asian or Asian British

Some women were concerned about the lack of clarity and the changing nature of advice.

*"I wanted it but then got pregnant. In that time, they keep moving the goal posts. Plus it's too soon to have a great understanding and statistics."* 40 years, White British

*"When I was pregnant—at the start it wasn't advised for pregnant women to have. Then half way through that was changed to we should. In a matter of months with little evidence as to why. I chose to decline it. I worked through the pandemic as a funeral coordinator and on private ambulance. Covid was sadly wiping out like flu does but earlier. I personally did not see people my age. Or middle aged passing of covid. So I am not happy to take a vaccine that could put myself and my baby at risk or my future."* 30 years, White British

**Lack of trust in COVID-19 vaccines.** Many women mentioned that they did not trust the vaccine or had any confidence in its necessity, safety or effectiveness. Some women believed that the vaccines did not prevent people from contracting or spreading the infection. Further, a few women had lower perception of risk from COVID-19 due to being younger and healthier.

*"I don't trust."* Age and ethnicity not disclosed

*"The infections have now risen so high when so many people are jabbed. It does not seem to be working."* 32 years, White British

*"It doesn't stop you catching it or spreading it—it only potentially reduces the symptoms and as a young fit healthy woman, I'd rather not take the risk of injecting the unknown into my body. If the vaccine prevented you getting it or spreading then I'd reconsider as it's helping others but right now, the only person it's helping is me and weighing up the risks, I'm happy not getting it for now."* 28 years, White Other

*"I trust the NHS information however I think the governments' message is wrong and misleading. Forcing young healthy people to get a vaccine when they probably wouldn't end up on hospital and they can still spread it is so wrong! Right now, as the vaccine stands, it's a personal choice which only affects the person getting the dose and nobody else. If I chose not to get it, I'm not putting anyone else at a greater risk then someone who is vaccinated as we can both spread it."* 28 years, White Other

**Concerns about the impact of vaccines on fertility.** Some women specifically mentioned the effect of the vaccine on their fertility.

*"Most concerns are around the impact on fertility."* 39 years, White British

*"My doctor initially advised against and after a miscarriage last year I've been too scared to have it."* 40 years, White Other

*"It is still in a trial phase therefore it is not known what side effects can happen to me or baby."* White British

## Information sources for COVID-19 vaccines

Common sources of information included patients' GPs and midwives, health-related websites, mainstream news organisations, patients' families and friends and social media respectively **Table 2**. The single most trusted sources of information were the patients' local GP or Midwife (n = 186, 42.8%) and specific health related websites (n = 170, 39.3%) **Table 2** and **Fig 2**. Other

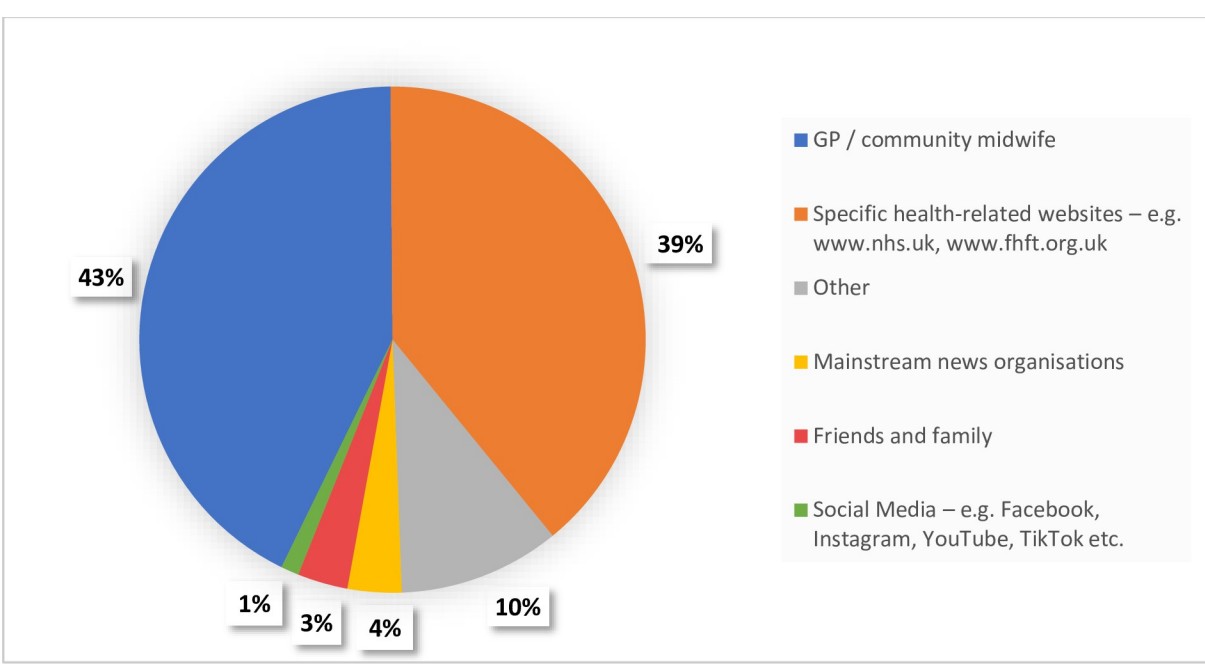

**Fig 2. Single most trusted sources of information for COVID-19 vaccines among pregnant women (n = 435).**

sources of information included Google searches, workplace, research and news from outside the UK, health websites such as Royal College of Obstetrics and Gynaecology, other healthcare professionals such as an obstetrician, nurse or hospital doctor and information directly from vaccine manufacturers. Some patients responded that they did not trust any source and relied on their own opinion/gut instinct or 'independent search' and some reported that it was a "guessing game".

## Discussion

### Main findings

A third of these pregnant women had not been vaccinated against COVID-19. The most common reasons were the possible adverse effects of the vaccine on the unborn baby and future pregnancies, lack of information about the vaccines' safety and effectiveness, concerns about the effects of the vaccine on the pregnant mother, concerns about the effects on fertility and lack of trust in COVID-19 vaccines.

Common sources of information about COVID-19 vaccination were the pregnant women's GP and midwife, health-related websites and mainstream news organisations. The single most trusted source of information for patients was their GP and midwife (42.8%).

### Interpretation in light of other evidence

This study confirms previous findings in the UK population and internationally that showed lower COVID-19 vaccine uptake amongst pregnant women than women in the general population [5, 7]. However, our sample was predominantly White British. It had fewer ethnic minority patients, which may explain higher vaccine uptake than previous studies [2, 7, 17, 24]. Only a quarter of those included were from ethnic minority groups. These groups are known to have lower participation in research [25]. The causes of low participation in research are likely to include personal and structural barriers, including lack of trust, language and

cultural factors, as well as socio-economic and systemic drivers [25]. Vaccine hesitancy amongst pregnant women is not unique to COVID-19 vaccination. Data from the Primary Care Networks serving the pregnant population of the two hospitals (Frimley and Wexham Park) show that the uptake of pertussis vaccine during pregnancy was approximately 53% (n = 2598/4896) between September 2021 to February 2022 (six months). A multi-methods study in the UK in 2022 showed that pregnant women who had not been vaccinated against pertussis were about four times (p-value < 0.0005) more likely to reject the COVID-19 vaccine [3].

The sample in our study differs from other samples in the literature [26, 27] as most women (64%) were getting COVID-19 information directly from a healthcare professional (i.e., GP and midwife), and this was the preferred source of information for almost half the sample (43%). It is possible that those who trust the healthcare professionals and the healthcare system were more likely to complete the survey. Midwives are regarded as trusted sources of information along with GPs in this survey. However, it is significant to highlight that midwives were reported to have lower COVID-19 vaccine uptake in a cohort study in 2021 [28]. Further, a preprint study said 4-times lower COVID-19 vaccination rates amongst black midwives compared to White midwives in London [29]. Therefore, some vaccine-hesitant midwives could negatively influence pregnant women's vaccination.

Around a quarter of the sample (23%) reported using social media for information about COVID-19, which may influence vaccine hesitancy. The spread of misinformation and disinformation, particularly through social media, has been highlighted as a major risk to ending the COVID-19 pandemic [30]. Of particular relevance to the sample is the prevalent misinformation on the theme of fertility. This misinformation can be presented in shocking terms, as demonstrated in these extracts from social media: "Big Pharma whistle-blower: '97% of corona vaccine recipients will become infertile'" and "The COVID-19 vaccine will make people infertile and is an attempt to reduce the population, particularly targeted at people from ethnic minority communities" [31, 32]. Finally, lack of clear communication of the reasons behind the change in advice by JCVI, within a short time, may have contributed to vaccine hesitancy amongst pregnant women and may have raised questions about the risks and benefits of the vaccine for this cohort. Furthermore, conflicting and inconsistent messaging is one of the drivers of vaccine hesitancy [17, 18].

## Strengths and limitations

This is the first UK survey, to our knowledge, among ethnically diverse pregnant women in a secondary care setting about the reasons for low uptake of COVID-19 vaccines and pregnant women's sources of information for COVID-19 vaccination. This was a large survey among pregnant women of a range of ages, ethnicities, languages, comorbidities, and gestations. It was conducted in a socio-economically diverse part of Southeast England. The survey inquired about pregnant women's sources of information and reasons for declining vaccination. We analysed both quantitative and qualitative (free text) data.

The main limitation is that we were unable to collect reliable information on overall response rate as the survey was completed in different ways by the respondents. We estimate that over half (63%) completed the survey during the consultant clinics at the Wexham Park and a similar number completed when active recruitment was possible in Frimley Park Hospital. A smaller number of women completed the survey as results of seeing posters in the clinic waiting rooms and toilet facilities. Clinicians reported that around a third declined to participate. The main reasons given were lack of time and interest. Active recruitment and data collection were not always possible due to workload pressures and COVID-19 restrictions.

Another weakness is that the survey was carried out in two hospitals in Southern England and may not apply to pregnant women in other areas or attending community-based antenatal clinics.

Depending on clinician capacity, the survey was undertaken amongst 2666 relatively high-risk pregnancies registered across both sites over six months (2021–2022). It is likely that some women were already anxious about their pregnancy and, therefore, would be less likely to engage with vaccination if they perceived an increased risk of complications. The survey was only conducted in the English language, which may be the reason for the inclusion of fewer ethnic minority patients compared to White British. Whilst White British make up 36% of booked patients; they contributed to approximately 50% of patients' responses in one hospital. Finally, we did not ask vaccinated women whether they were vaccinated during or before their pregnancy. This would be crucial as fully vaccinated women who had the first two doses of the vaccine before getting pregnant might have declined further booster doses during pregnancy.

## Recommendations to increase COVID-19 vaccine uptake in pregnancy

These recommendations are based on the findings of this survey and existing evidence. Some authors (MSR, PO) have previously proposed the five Cs of vaccine hesitancy: confidence, complacency, convenience, communication and context (sociodemographic) [18, 33]. Further practical recommendations are provided in **Table 3**.

1. Confidence in the safety and effectiveness of the vaccine for mother and unborn baby are strong predictors of uptake. Therefore, it is essential to engage in transparent dialogue that respects pregnant women's concerns and addresses information needs.

**Table 3. Recommendations to improve COVID-19 vaccination uptake among pregnant women.**

| Factors influencing uptake | Specific concerns among pregnant women | Opportunities to address this with pregnant and postnatal women | Considerations & Resources for practice: |
|---|---|---|---|
| Concerns about safety and long-term effects of COVID-19 vaccines on baby and mother | Concerns about speed of vaccine development and vaccine roll out | Explain the biological principles of vaccines and vaccination in pregnancy, the rigorous vaccination approval process, international collaboration, significant resource allocation during the pandemic | • Primary and Secondary care to ensure consistent messaging |
| | | | • Use of trusted sources of information such as GP, midwives and obstetricians |
| | | | • Utilise local community hubs, through in-reach community vaccination programme delivery |
| Lack of information and evidence on vaccines' safety and effectiveness | Mixed messaging from government and healthcare bodies | Understand previous misinformation exposure in order to address concerns directly | • Training and updates for relevant staff on safety and efficacy of the vaccine. |
| | | Address information needs and provide data in an accessible manner | • Effective use of existing communication tools to explain risks and benefits |
| | | | • For those unvaccinated and in at-risk categories, opportunity for discussion with healthcare professional who are fully informed themselves as they will likely be perceived as role models |
| Lack of confidence and trust in COVID-19 vaccines | Mistrust of government and healthcare professionals | Providing information from trusted Healthcare Professionals | • Building trust through healthcare professionals who have established rapport through long term relationship-based care |
| | | Consistent, transparent and honest explanation about the evolving nature of evidence and uncertainty | • Address the socio-economic and historic causes of mistrust especially in some ethnic minority groups |
| Concerns about the impact of vaccines on fertility | Exposure to misinformation about the impact of vaccines on fertility | Transparent information about risks and benefits of vaccination and explanation of data about vaccines' safety | • Appoint Vaccine Champions to establish transparent dialogues with perinatal women |
| | | Communication through appropriate social media. | • Appropriate use of social media to disseminate tailored information from trusted sources such as GPs and midwives |

2. Communication: vaccine communication should be undertaken by practitioners who are trusted sources of information (i.e., GP, midwife, obstetrician) during routine antenatal appointments in both primary and secondary care. However, it is crucial that staff, including midwives, are trained in rapport building and engaging in dialogue with pregnant women about the risks and benefits of the vaccine to mother and baby, countering misinformation and conspiracy theories. Furthermore, vaccine communication training should be prioritised amongst all relevant staff.

3. Complacency: perceived lower risk of COVID-19 infection and severe disease is a strong predictor of low uptake. Therefore, complacency should be addressed by repeated risk communication that facilitates informed decision-making by women. Additionally, some staff should be delegated and trained as vaccine champions who would be available to provide comprehensive and accurate information for pregnant women.

4. Convenient, tailored pathways for vaccination should be available. Further, providers should seek to understand the local community's specific barriers and facilitators and address those sensitively, ensuring that underserved groups can be reached.

5. Context: empathetic dialogue using motivational interviewing skills with vaccine-hesitant pregnant women, especially those from ethnic minority groups with higher vaccine hesitancy, should be used to acknowledge their fears of harming their unborn child.

## Conclusions

This large quality improvement survey amongst a diverse group of pregnant women highlights barriers and facilitators to COVID-19 vaccination in pregnancy. As a result, we have already appointed vaccine champions at our hospital. The recommendations based on this study could improve both COVID-19 vaccination uptake and other routine immunisations, such as pertussis and influenza vaccines, during pregnancy.

## Supporting information

**S1 Fig. Anonymous questionnaire among pregnant women in Frimley NHS Trust.**
(PDF)

## Acknowledgments

We thank all participants for sharing their views and experiences. We are immensely grateful to all the staff at Frimley Park and Wexham Park Hospitals for their help and assistance with this survey. This would not have been possible without their kindness and generosity.

## Author Contributions

**Conceptualization:** Fatima Husain.

**Data curation:** Fatima Husain, Veronica R. Powys, Eleanor White, Roxanne Jones.

**Formal analysis:** Lucy P. Goldsmith.

**Investigation:** Fatima Husain.

**Methodology:** Fatima Husain, Veronica R. Powys, Eleanor White, Roxanne Jones.

**Project administration:** Veronica R. Powys, Eleanor White, Roxanne Jones.

**Resources:** Veronica R. Powys, Eleanor White.

**Supervision:** Fatima Husain, Mohammad Sharif Razai.

**Validation:** Eleanor White, Paul T. Heath, Pippa Oakeshott.

**Writing – original draft:** Fatima Husain, Veronica R. Powys, Eleanor White.

**Writing – review & editing:** Fatima Husain, Veronica R. Powys, Eleanor White, Roxanne Jones, Lucy P. Goldsmith, Paul T. Heath, Pippa Oakeshott, Mohammad Sharif Razai.

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
