## [Decision Letter · Decision Letter 0]

9 May 2022

PONE-D-22-11364COVID-19 vaccination uptake in 441 socially and ethnically diverse pregnant women: a mixed-method surveyPLOS ONE

Dear Dr. Razai,

Thank you for submitting your manuscript to PLOS ONE. After careful consideration, we feel that it has merit but does not fully meet PLOS ONE’s publication criteria as it currently stands. Therefore, we invite you to submit a revised version of the manuscript that addresses the points raised during the review process.

Specifically these areas are in need of revision, clarification, and improvement:  

Design of the study should be revised

Introduction section should be expanded      

Results should be improved:

The numbers of women attending the hospital and clinics during the period of the study.

The percentage of women who did not respond to invitation to participate in the study

The percentage of women who did not give their consent to participate  

Discussion section should be more robust:    

The rate of other pregnancy vaccines uptake in the study hospitals for comparison

Comparisons with recent similar works across continents and comparison with national vaccine uptake

Recommendations

We look forward to receiving your revised manuscript.

Kind regards,

Forough Mortazavi

Academic Editor

PLOS ONE

Journal Requirements:

" MSR is funded with a NIHR In-practice Fellowship (NIHR 302007). "

" The authors received no specific funding for this work"

Reviewers' comments:

Reviewer's Responses to Questions

**Comments to the Author**

1. Is the manuscript technically sound, and do the data support the conclusions?

Reviewer #1: Partly

Reviewer #2: Yes

2. Has the statistical analysis been performed appropriately and rigorously? 

Reviewer #1: Yes

Reviewer #2: Yes

3. Have the authors made all data underlying the findings in their manuscript fully available?

Reviewer #1: Yes

Reviewer #2: Yes

4. Is the manuscript presented in an intelligible fashion and written in standard English?

Reviewer #1: Yes

Reviewer #2: Yes

5. Review Comments to the Author

Reviewer #1: Although there have been a number of national surveys investigating uptake of COVID-19 among pregnant women, it is important to understand the local situation with respect to vaccines uptake locally. This is a clearly written paper describing a study set in two English hospitals.

I would suggest expanding the introduction section to include some detail about the original recommendation re COVID-19 vaccine in pregnancy ie that it was not recommended routinely https://www.gov.uk/government/publications/priority-groups-for-coronavirus-covid-19-vaccination-advice-from-the-jcvi-30-december-2020/joint-committee-on-vaccination-and-immunisation-advice-on-priority-groups-for-covid-19-vaccination-30-december-2020

This is important to set the context for the reader who may not be familiar with the change in advice (eg an International reader) and also because some women refer to this as their reason for not accepting the vaccine.

Due to the methods used it is understandable that no response rate could be calculated. However some estimation could be made as the numbers of women attending during the period of the study must be known, equally information on ethnicity generally in this population and of chronic health conditions so that some comment on representativeness could be made.

My main comments are with the description of the type of study. This is ostensibly a survey and although open ended responses were 'analysed thematically' and quotes included, I would suggest this does not make this a mixed methods study, nor the open ended data 'qualitative' in which you would expect in depth interviews to have been conducted in addition to the survey.

In the section on information sources it is reported that the single most trusted sources of information were the patient's local GP or midwife. this is a common finding in studies of vaccination uptake. On this basis the authors then go on to recommend that trusted sources of information should be used to address concerns about vaccine safety and long term vaccine safety. While this is not unreasonable in itself, I would suggest that a trusted sources does not necessarily equal an accurate source. Indeed, it has been shown that midwives themselves often do not accept COVID-19 vaccine in fact in the SIREN study, they were the health staff group with the highest non acceptance (https://www.sciencedirect.com/science/article/pii/S014067362100790X). another study (not peer reviewed) found important differences in uptake among midwives with a minority ethnic background (https://assets.researchsquare.com/files/rs-646142/v1/f151867e-9f92-48eb-b554-bea59db31657.pdf?c=1631888999) and so I would be inclined to explore further with midwives, their views about COVID- 19 vaccine. Training of staff is not mentioned in the recommendations, which should be a priority. It would be useful to look at the data in more detail to explore this issue perhaps a cross tab of 'most trusted source' x vaccine uptake - though probably not possible as GP and community midwife are one category.

Are data available on uptake of other pregnancy vaccines in the study hospitals for comparison? Skirrow et al (ref 3) found an association between non acceptance of pertussis vaccine and COVID vaccine among pregnant women.

in addition a comparison with national uptake would be useful (53.7% on 24th March 2022)

A few minor details:

Page 5 - last but last line 'by an experienced statistician' is not needed

Page 6 - missing full stop after syndrome

Page 9 - second para - typo heath

Reviewer #2: The study is relevant as it is important to explore the views of special groups of people (in this case pregnant women) towards COVID-19 vaccination. Find some comments below:

Abstract

Kindly rephrase the method by removing "we". It can be stated as "Consented pregnant women completed thr electronic survey using a QR code or on paper. Data were summarised descriptively and text responses were thematically analysed."

Discussion

Discussion section should be more robust. Focusing on the objectives, vaccination uptake by pregnant women facilitators and barriers with factors responsible for these can be comprehensively discussed. Sugnificance of ethic diversity can be discussed. Discussion of present findiwith literature summarised under introduction can also be explained.

6. PLOS authors have the option to publish the peer review history of their article (what does this mean?). If published, this will include your full peer review and any attached files.

Reviewer #1: No

Reviewer #2: No

---

## [Author Response · Author response to Decision Letter 0]

11 Jun 2022

We thank the reviewers and editors for their thoughtful comments which we think have improved the paper. Please find below our response to the points raised. We have responded to specific reviewer and editor comments in a separate file attached along with manuscript, manuscript with track changes and supplementary figure S1.

---

## [Decision Letter · Decision Letter 1]

8 Jul 2022

COVID-19 vaccination uptake in 441 socially and ethnically diverse pregnant women

PONE-D-22-11364R1

Dear Dr. Razai,

We’re pleased to inform you that your manuscript has been judged scientifically suitable for publication and will be formally accepted for publication once it meets all outstanding technical requirements.

Kind regards,

Forough Mortazavi

Academic Editor

PLOS ONE

Additional Editor Comments (optional):

Reviewers' comments:

Reviewer's Responses to Questions

**Comments to the Author**

1. If the authors have adequately addressed your comments raised in a previous round of review and you feel that this manuscript is now acceptable for publication, you may indicate that here to bypass the “Comments to the Author” section, enter your conflict of interest statement in the “Confidential to Editor” section, and submit your "Accept" recommendation.

Reviewer #1: All comments have been addressed

2. Is the manuscript technically sound, and do the data support the conclusions?

Reviewer #1: Yes

3. Has the statistical analysis been performed appropriately and rigorously? 

Reviewer #1: Yes

4. Have the authors made all data underlying the findings in their manuscript fully available?

Reviewer #1: Yes

5. Is the manuscript presented in an intelligible fashion and written in standard English?

Reviewer #1: Yes

6. Review Comments to the Author

Reviewer #1: (No Response)

7. PLOS authors have the option to publish the peer review history of their article (what does this mean?). If published, this will include your full peer review and any attached files.

Reviewer #1: No

---

## [Editor Report · Acceptance letter]

21 Jul 2022

PONE-D-22-11364R1 

COVID-19 vaccination uptake in 441 socially and ethnically diverse pregnant women 

Dear Dr. Razai:

I'm pleased to inform you that your manuscript has been deemed suitable for publication in PLOS ONE. Congratulations! Your manuscript is now with our production department. 

Kind regards, 

on behalf of

Dr. Forough Mortazavi 

Academic Editor

PLOS ONE